# An Update on COVID-19 Vaccine Induced Thrombotic Thrombocytopenia Syndrome and Some Management Recommendations

**DOI:** 10.3390/molecules26165004

**Published:** 2021-08-18

**Authors:** Amin Islam, Mohammed Sheraz Bashir, Kevin Joyce, Harunor Rashid, Ismail Laher, Shereen Elshazly

**Affiliations:** 1Department of Haematology, Mid & South Essex University Hospital NHS Foundation Trust, Prittlewell Chase, Westcliff-on-Sea SS0 0RY, UK; mohammed.bashir3@nhs.net (M.S.B.); kevin.joyce@nhs.net (K.J.); shereen.elshazly@nhs.net (S.E.); 2Department of Haematology, Queen Mary University of London, Mile End Road, London E1 3NS, UK; 3National Centre for Immunisation Research and Surveillance(NCIRS) Kids Research, The Children’s Hospital at Westmead, Westmead, NSW 2145, Australia; harunor.rashid@health.nsw.gov.au; 4The Discipline of Child and Adolescent Health, Faculty of Medicine and Health, The University of Sydney, Camperdown, NSW 2145, Australia; 5Department of Anesthesiology, Pharmacology & Therapeutics, Faculty of Medicine, The University of British Colombia, Vancouver, BC V6T 1Z3, Canada; ismail.laher@ubc.ca; 6Adult Haemato-Oncology Unit, Faculty of Medicne, Ainshams University, Cairo 11566, Egypt

**Keywords:** COVID-19, COVID-19 vaccine, disseminated intravascular coagulation, heparin induced thrombocytopenia, platelet factor 4, thrombosis, thrombotic thrombo-cytopenic purpura, vaccine induced thrombocytopenia and thrombosis

## Abstract

The thrombotic thrombocytopenia syndrome (TTS), a complication of COVID-19 vaccines, involves thrombosis (often cerebral venous sinus thrombosis) and thrombocytopenia with occasional pulmonary embolism and arterial ischemia. TTS appears to mostly affect females aged between 20 and 50 years old, with no predisposing risk factors conclusively identified so far. Cases are characterized by thrombocytopenia, higher levels of D-dimers than commonly observed in venous thromboembolic events, inexplicably low fibrinogen levels and worsening thrombosis. Hyper fibrinolysis associated with bleeding can also occur. Antibodies that bind platelet factor 4, similar to those associated with heparin-induced thrombocytopenia, have also been identified but in the absence of patient exposure to heparin treatment. A number of countries have now suspended the use of adenovirus-vectored vaccines for younger individuals. The prevailing opinion of most experts is that the risk of developing COVID-19 disease, including thrombosis, far exceeds the extremely low risk of TTS associated with highly efficacious vaccines. Mass vaccination should continue but with caution. Vaccines that are more likely to cause TTS (e.g., Vaxzevria manufactured by AstraZeneca) should be avoided in younger patients for whom an alternative vaccine is available.

## 1. Background

A new deadly virus of the coronavirus family was first identified in December 2019 and named SARS-2-CoV-2; this virus caused severe acute respiratory syndrome and is now known as COVID-19. Patients presented with variable symptoms, ranging from asymptomatic carriers to life-threatening/changing consequences. Several vaccines have been developed and are currently being used to reduce disease incidence and mortality in many countries. Lately, rare but life-threatening events such as thrombosis with thrombocytopenia syndrome (TTS) (also called VITT—vaccine induced thrombocytopenia and thrombosis) have been reported with some COVID-19 vaccines. Recent reviews of TTS following COVID-19 vaccinations have not included clinical management guidelines [1,2]. To this end, this review summarizes the available data on the pathophysiology of COVID-19 and thrombosis, the different types of vaccines used to prevent COVID-19, the proposed mechanisms of TTS and some clinical management recommendations.

## 2. COVID-19 and Thrombosis

Approximately 40–45% of SARS-2-CoV-2 infected individuals are asymptomatic [3], while 14% develop severe illness, and 5% are critically unwell [4]. The clinical course of COVID-19 can be severe and sometimes associated with complications such as venous thromboembolism events (VTEs), severe inflammatory response syndrome, acute respiratory distress syndrome and multi-organ dysfunction syndromes, particularly in the elderly or those with co-morbidities such as diabetes mellitus, renal disease and cardiovascular conditions [5]. More than one-fifth of all patients with COVID-19 develop VTEs [6].

The pathophysiology of COVID-19 associated thrombosis is not completely understood. The associated hyper-inflammatory response produced by COVID-19 is inter-woven with other pathways, involving pro-inflammatory mediators, endothelial damage and direct invasion of cells such as type 2 pneumocytes, and coagulopathies such as disseminated intravascular coagulation (DIC) [7].

Platelets contain chemokines, chemotactic factors, various adhesion molecules, co-stimulatory molecules in their membranes and granules to support their role in haemostasis and immunomodulation. Platelets trigger blood coagulation and inflammation and initiate innate immune responses through the expression of Toll-like receptors (TLRs) to release inflammatory cytokines, trigger adaptive immune responses and activate T cells through the expression of key costimulatory molecules and major histocompatibility complex (MHC) molecules. Platelets release large amounts of extracellular vesicles which can interact with multiple immune mediators. Therefore, the function of platelets extends beyond aggregation, and the interaction with haemostasis, inflammation and the immune response results in the amplification of the body’s defence mechanisms [8].

An abnormal viral coagulopathy typically results in a pro-thrombotic phenotype that can cause micro-thrombi and macro-thrombi, resulting in both venous and arterial occlusions [9]. There have also been rare isolated cases of thrombotic thrombocytopenic purpura (TTP) [10]. Thrombotic complications were reported early in our understanding of viral induced clinical syndromes of COVID-19 infection, and they are now a leading cause of death during the COVID-19 pandemic.

Early research identified various laboratory parameters to help stratify thrombotic risk, resulting in many hospitals in the United Kingdom (UK) adopting a COVID-19 test panel consisting of D-dimer tests, clotting screen, fibrinogen, serum ferritin, lactate dehydrogenase and troponin levels. This enabled VTE prophylaxis to be adapted to individual patient risk, with low molecular weight heparin used as an anticoagulant for inpatients and continued post discharge for periods of up to 12 weeks. Oral anticoagulants are also useful in some patients post discharge [11].

Thrombocytopenia occurred in almost one in three COVID-19 (31.6%) infected inpatients and at a greater rate in individuals with severe COVID-19 (57.7%) [12]. The majority of patients presented with mild thrombocytopenia, with some cases of severe thrombocytopenia requiring a careful balance between bleeding and thrombotic risk. Several pathophysiological mechanisms have been postulated for thrombocytopenia in such cases. These include bone marrow suppression that is somewhat similar to that observed in sepsis; there is likely direct infection of the marrow by SARS-2-CoV-2, affecting megakaryocytic and haematopoietic precursor cell function and synthesis. Other mechanisms include reduced thrombopoietin synthesis in the liver, generation of micro-thrombi with subsequent platelet consumption and, finally, immune thrombocytopenia (ITP) resulting in peripheral platelet destruction [13]. Infection with COVID-19 and injury to the type II alveolar cells results in increased expression of p53, suppression of uPA and uPAR and increases in PAI-1. In addition, the type II alveolar cells are also the source of surfactant; viral infection diminishes the release of surfactant. There is a temporal relationship between COVID-19 infection and fibrinolysis. In the acute phase of the infection, the inflow of fibrinogen and coagulation factors result in fibrin deposition and hyaline membrane formation. Acute inflammatory cytokines consisting of IL-1, IL-6 and IL-17A upregulates plasminogen activator inhibitor-1 (PAI-1) and suppresses the expression of urokinase-type plasminogen activator (uPA) and urokinase-type plasminogen activator receptor (uPAR). Reduced surfactant levels activate the p53 pathway, resulting in increases in PAI-1 and decreases in uPA and uPAR. The fibrinolytic balance is then shifted to a hypofibrinolytic state, which stimulates fibrin deposition, hyaline membrane formation and microvascular thrombosis [14,15,16].

## 3. Vaccinations

The World Health Organization (WHO) estimates that over 200 million people have so far been diagnosed with COVID-19, with over 4 million deaths worldwide [17]. The identification of this virus stimulated a global race to develop a vaccine. The UK was the first country to start a mass vaccination campaign with Comirnaty (the Pfizer-BioNTech vaccine, Mainz, Germany) for COVID-19 in December 2020; this was then quickly followed by approval of Vaxzevria (AstraZeneca, Oxford, UK). The Moderna COVID-19 vaccine was approved in January 2021 [18]; other vaccines include Sputnik V (Gamaleya Institute, Moscow, Russia), BBBP-CorV (SinoPharm, Shanghai, China), Janssen COVID-19 vaccine (Johnson & Johnson, New Brunswick, NJ, USA) and Covovax (Novavax, Gaithersburg, MD, USA).

There are currently at least nine different technology platforms used to create vaccines against COVID-19. Most vaccine candidates undergoing clinical trials focus on the coronavirus spike protein and its variants as the primary antigen of COVID-19 infection. These methods involve nucleic acid technologies (nucleoside-modified messenger RNA and DNA), non-replicating viral vectors, peptides, recombinant proteins, live attenuated viruses and inactivated viruses [19].

The vaccine types are listed as follows:**RNA vaccines**: An RNA vaccine contains RNA which, when introduced into a tissue, acts as messenger RNA (mRNA) in order to cause cells to synthesize spike proteins and stimulate an adaptive immune response. RNA vaccines often, but not always, use the nucleoside-modified messenger RNA. The delivery of mRNA is achieved by using lipid nanoparticles to protect the RNA strands and to facilitate their absorption into cells. The best-known examples are the Comirnaty and the Moderna COVID-19 vaccines [20].**Adenovirus vector vaccines:** These vaccines are examples of non-replicating viral vector vaccines that use an adenovirus shell containing DNA encoding a SARS-CoV-2 protein. These vaccines are non-replicating and produce only the antigens that elicit a systemic immune response [21]. Examples of this group include Vaxzevria, Sputnik V [22], Convidecia (CanSino Biologics, Tianjin, China) and Janssen COVID-19 vaccines [23].**Inactivated viral vaccines:** These vaccines include viral particles grown in culture and subsequently inactivated toward non-pathogenic particles with immunogenic properties. Vaccines of this type include CoronaVac (Sinovac Biotech, Beijing, China); BBIBP-CorV and WIBP-CorV (Sinopharm, Beijing, China); Covaxin (Bharat Biotech, Hyderabad, India); and CoviVac (Chumakov Centre, Moscow, Russia) [24].

Despite recent surges of COVID-19 in some countries (e.g., Israel, UK) that have been fueled by more transmissible variants, mass vaccination programs have nonetheless been generally successful with fewer deaths reported after implementation of the vaccination program. A single inoculation with Comirnaty or Vaxzevria in the UK resulted in an 85% reduction in hospitalization from COVID-19 infections [13]. It is estimated that programs in the UK (Comirnaty and Vaxzevria inoculations) have prevented at least 10,400 deaths as of March 2021 [25]. The number of deaths in the UK within 28 days of a positive COVID-19 test has been steadily falling since mid-January 2021.

Data from the official UK government dashboard suggests that deaths from coronavirus in those who are 80 years or older in England fell by 62% between 24 January and 12 February 2021 [26]. This compares to a drop of 47% in those aged between 20 and 64 years old and of 51% in those between aged 65 and 79 years old. Further evidence comes from Scotland which has seen deaths from COVID-19 falling in all locations, with the fastest decreases occurring in long-term care homes where deaths fell by 62% in the three weeks leading to 14 February 2021, approaching a level last observed near the end of October 2020. Older residents in care homes were prioritized when the vaccination program began. The report from the National Records of Scotland shows that the number of deaths in those aged 85 and over has fallen by 45%, which is steeper than in younger age groups [27].

## 4. Vaccines and Serious Adverse Events

Adverse events have been reported with most of the COVID-19 vaccines; these are typically mild (pain at the injection site, myalgia, headaches, fatigue and tiredness) and usually resolved within a few days. The most frequent adverse reactions in adolescents 12 to 15 years of age were injection site pain (>90%), fatigue and headache (>70%), myalgia and chills (>40%) and arthralgia and pyrexia (>20%). These are the data published and regularly updated on UK government central adverse drugs reactions reporting systems [28].

However, some patients experienced rare side-effects that sometimes were fatal. The early adoption of Comirnaty as frequently associated with reports of anaphylaxis type reactions and some safety concerns were raised, resulting in some recommendations for increased monitoring of patients after receiving the injection and even avoiding Comirnaty in those with a history of allergic reactions. These measures substantially reduced the rates of serious adverse events.

Attention has now centred on VTEs that occur 5 to 24 days after the first inoculation with Vaxzevria. It is important to note that VTE is not an uncommon event and can occur naturally at any age. Even as new waves of COVID-19 emerge in many countries, apprehension remains relative to the safety and efficacy of Vaxzevria, with several countries in the European Union either entirely suspending inoculations or restricting its use to selected age groups [20,21]. For example, Austria suspended the use of a single batch (ABV5300) of vaccines pending further investigation [29], while the UK has advised that an alternative should be sought for those aged below 30 years, and Germany has suspended its use in those aged 60 years and under [30]. At the same time, these concerns led France and Germany to consider using Sputnik V [31]. The marketing authorization holder report concluded in its observed-to-expected analysis that the number of deep vein thrombosis (DVT) or pulmonary embolisms (PE) cases observed was in fact significantly lower than expected, suggesting no causal association between VTEs and Vaxzevria [32]. However, the pharmacovigilance risk assessment committee stressed caution in this interpretation due to concerns related to quality, sensitivity and appropriate stratification in the marketing authorization holder report. The recently approved Janssen COVID-19 vaccine (which also uses an adenovirus vector) includes VTEs in its risk management plan [33].

## 5. Thrombosis with Thrombocytopenia Syndrome

TTS is an extremely rare but increasingly recognized serious adverse event related to unusual sites of thromboembolism, such as cerebral venous sinus thrombosis (CVST) or abdominal thromboses (splanchnic, mesenteric or portal vein), all of which are associated with thrombocytopenia. ‘CVST with thrombocytopenia’ is a rare subtype of cerebrovascular accident, with an incidence of 5.0 per million in those receiving Vaxzevria and 4.1 per million in those receiving mRNA based vaccines, and the prevalence is three times greater in younger to middle aged women (mean age 35) [34,35,36,37]. An international study on cerebral venous and dural sinus thrombosis, as one of the largest prospective cohort studies on CVST, confirmed this gender bias for CVST. The presentation of CVST was initially described in case-controlled studies, but later confirmed in a meta-analysis (pooled odds ratio: 5.59) that proposed gender-specific risk factors such as oral contraceptives [38], hormone replacement therapies and pregnancy (including the post-partum period) [39]. This is further supported by epidemiological studies demonstrating a correlation between the use of oral contraceptive and CVST in younger to middle aged females [40]. The epidemiology of CVST has not been carefully studied, making it difficult to comprehend and compare the data in the context of the COVID-19 era.

At least 12 cases of CVST with thrombocytopenia have been reported in the USA from 2 March to 21 April 2021 [41], and case reports were also published from European countries [42,43]; starting in the first week of April 2021, with a total of 169 cases of CVST with thrombocytopenia reported among 34 million people vaccinated Vaxzevria in the European Economic Area (EEA) and the UK [44].

Recent estimates suggest a 100-fold increased risk of developing a CVST after being infected with COVID-19, with a third of cases occurring in those aged less than 30 years old [37]. A non-peer reviewed report identified a 30-fold increased risk of splanchnic vein thrombosis in recipients of mRNA based vaccines such as the Comirnaty or Moderna COVID-19 vaccines (1 per 1.6 million for Vaxzevria recipients vs. 1 per 44.9 million for mRNA vaccines) [37]. It is important to note that the report focusses on cases of CVST and splanchnic thrombosis without associated thrombocytopenia.

A Danish population study of mostly female healthcare workers who received Vaxzevria identified no clear causal association between the vaccine and blood clots [45]. Records on the time course of VTEs in relation to the vaccine administration are unknown. It is also unclear if the patients who developed VTE were infected with COVID-19 (asymptomatic or otherwise) prior to or immediately before immunity developed [45]. It is difficult to apply data from this Danish study to other regions or to gauge the prevalence of different COVID-19 strains or to determine the viral pathogenicity and associated secondary complications. The underlying co-morbidities or pre-existing risk factors such as a history of previous VTEs or thrombophilia were also not considered. The rapid spread of the COVID-19 pandemic limited the Danish study to only pre-COVID-19 era data for the comparisons of VTEs. Importantly, the study reinforced early concerns about the unusual sites of thrombosis with thrombocytopenia shortly after receiving Vaxzevria.

The overall interim recommendation and shared agreement between the WHO, European Medicines Agency (EMA) and MHRA is that the intended benefits of receiving a vaccine for COVID-19 still far outweighs its rare side-effects. More recently, TTS was reported in six patients receiving the Janssen vaccine [46]. Vaxzevria and the Janssen vaccine both consist of recombinant adenoviral vectors based on a chimpanzee adenovirus (Vaxzevria) or a human adenovirus (Janssen vaccine) that encodes the SARS-CoV-2 spike protein immunogen. While a few individuals receiving the Moderna COVID-19 vaccine lipid nanoparticle encapsulated mRNA vaccine were diagnosed with CVST, the FDA reports that there are currently no reports of patients who developed TTS after being vaccinated with Comirnaty or the Moderna COVID-19 vaccine. The striking clinical similarities of TTS and heparin-induced thrombocytopenia (HIT) and the uniformly positive platelet factor 4 (PF4)-heparin enzyme-linked immunosorbent assays (ELISAs) in these cases may be due to circulating PF4-reactive antibodies that can directly activate platelets in the absence of heparin. It is important to note that non-ELISA-based commercial assays (e.g., HemosIL AcuStar HIT-IgG, HemosIL HIT-Ab, Diamed PaGIA gel and STic Expert assays) are weakly sensitive for anti-PF4 antibodies in samples from patients with suspected TTS [47]. Intravenous immune globulin and a monoclonal antibody to the Fc receptor can block platelet activation by these antibodies, at least in vitro. These clinical and laboratory features are similar to rare cases of an HIT-like syndrome previously described after some medications or infections in patients not receiving heparin [48].

## 6. Possible Pathophysiology of TTS

Antibody-mediated thrombotic thrombocytopenia during COVID-19 is presumed to be an autoimmune reaction induced by SARS-CoV-2. The high incidence of thrombotic thromboembolic events during severe COVID-19 results in the frequent administration of heparin in affected patients [49]. HITT is a possible cause when thrombocytopenia is associated with thrombosis in this setting [50]. Several studies report the presence of anti-PF4/heparin antibodies in COVID-19 patients, these antibodies can also be found without any history of heparin administration [51]. Furthermore, these antibodies do not always activate platelets in the presence of heparin/PF4 complexes [52], although they can do so in presence of PF4 alone [53], suggesting that their production is likely unrelated to HIT [54]. Related to this notion is that IgG antibodies in the serum of severe cases of COVID-19 infections induce platelet apoptosis and procoagulant activity via FcγRIIA (CD32) receptor-dependent mechanisms [55]. The antigenic specificity of these antibodies is unclear, although it is likely that at least some of them are directed against PF4.

The model we support is based on the hyperactivation of platelets during COVID-19, which results in the release of PF4 into the circulation [56]. Circulating PF4 forms complexes with endogenous polyanionic proteoglycans released by damaged endothelial cells. Syndecan-1 and endocan are potential proteoglycans candidates as their serum levels are increased in severely ill COVID-19 patients in association with other markers of endothelial injury [57]. Complexes formed between PF4 and endothelial cell-derived polyanionic proteoglycans can then stimulate extra follicular B cells that produce anti-PF4 antibodies, as suggested by previous reports that autoimmune responses elicited by extra follicular B cells may be involved in the pathophysiology of severe COVID-19 [58].

A recent study by Kowarz and colleagues suggested a different slicing mechanism for spike open reading frame in adeno vector vaccines, which results in soluble spike variants that can initiate severe side effects when binding to ACE2-expressing vascular endothelial cells. They compared this phenomenon to thromboembolic events caused by spike protein encoded by the SARS-CoV-2 virus and termed this possible mechanism as the “Vaccine-Induced COVID-19 Mimicry” syndrome (VIC19M syndrome) [59].

Anti-phospholipid antibodies could also additionally contribute to platelet activation, as suggested by increases in anti-SARS-CoV-2 antibodies in other viral diseases [60]. The rare prothrombotic thrombocytopenic events following vaccination with Vaxzevria (~1 in 100 000 recipients) has a clinical presentation similar to HIT, suggesting that a vaccine-induced autoimmune response to PF4 may be plausible. Supporting this hypothesis is a recent study identifying platelet-activating anti-PF4 antibodies in the sera of patients suffering from unusual thrombotic events associated with thrombocytopenia within 4 to 16 days after receiving Vaxzevria [61]. The progression of this possible vaccine-induced anti-PF4 autoimmune response could be related to mechanisms similar to those for prothrombotic thrombocytopenia induced by the SARS-CoV-2 virus itself. Other possible mechanisms include adenoviral vector entry in megakaryocytes and the subsequent expression of spike protein on platelet surfaces and also direct platelet activation by the vector [62].

## 7. Heparin Induced Thrombocytopenia with Thrombosis

HIT with thrombosis is a severe prothrombotic condition that occurs in less than five percent of patients receiving intravenous unfractionated heparin, less commonly with low molecular weight heparin and usually between 4 and 10 days after initiation of treatment [63]. Thrombocytopenia is a hallmark of HIT with thrombosis, with platelet counts decreased by more than half in most patients. After exclusion of other causes of thrombocytopenia and HIT, a clinical diagnosis of HIT with thrombosis is established by immune enzymatic detection of circulating antibodies to PF4/heparin complexes, followed by a functional assay demonstrating platelet activation by the patient’s serum in the presence of heparin [63]. Risk factors for HIT with thrombosis include the following: (1) the duration of heparin therapy (>5 days); (2) the type (unfractionated heparin > low molecular weight heparin > fondaparinux) and dosage of heparin; (3) the indication for treatment (surgical and trauma patients at highest risk); and (4) gender (female > male) [64]. Thrombotic complications can develop in unusual locations such as cerebral venous sinuses [65].

The specificity of PF4 autoantibodies causing HIT with thrombosis was confirmed in studies demonstrating that the epitope recognized on PF4 tetramers was exposed after interaction with heparin or other long polyanions [54]. The injection of heparin releases PF4 [66], resulting in the assembly of PF4/heparin complexes which activate complement and bind circulating B lymphocytes in a complement-dependent manner [67]. B cells responsible for the synthesis of PF4 autoantibodies rapidly mount an IgG immune response following their first exposure to heparin [68]. B cells that produce anti-PF4 antibodies are present in healthy individuals in an anergic state in which there is an absence of an immune response to an antigen such as PF4. This B cell tolerance could be disturbed after exposure to heparin and also in some inflammatory conditions [69], where anti-PF4 IgG antibodies elicit thrombus formation and thrombocytopenia via multiple mechanisms (Figure 1)**.**

Immune complexes assembled with PF4 bound to heparin induce platelet activation and aggregation by crosslinking with FcγRIIa receptors [63]. The pathogenesis of HIT is shown in the schematic Figure 1. Anti-PF4 antibodies also activate the pro-coagulant activity of monocytes by cross-linking their FcγR1 receptors and increase the thrombotic activity of endothelial cells via recognition of PF4 firmly attached to surface proteoglycans [70]. Thrombocytopenia results from enhanced apoptosis and clearance of antibody-decorated platelets [71]. The similarities and differences between HIT with thrombosis and TTS are highlighted in Table 1.

## 8. Prevalence of Platelet Factor 4 Antibody in the General Population

An assessment of immunoassay results (11 studies; 860 subjects) on the prevalence of PF4/heparin antibody (IgG/M/A) in healthy subjects concluded that commercial immunoassays are able to detect PF4/heparin antibodies in 1.0–4.3% of healthy subjects [73]. Another prospective study measured PF4/heparin antibody levels in approximately 4000 blood donors by using a commercial enzyme-linked immunosorbent assay for initial findings and then repeated for confirmatory testing. Antibody levels were initially detected in 249 of 3795 donors (6.6%; 95% confidence interval [CI], 5.8–7.4%) and then confirmed in 163 of 3789 evaluable donors (4.3%; 95% CI, 3.7–5.0%). Of the 163 repeated positive samples, 116 (71.2%) were low positives, and 124 (76.1%) exhibited heparin-dependent binding. Predominant isotypes of intermediate to high seropositive samples (OD > 0.6) were IgG (20/39 (51%)), IgM (9/39 (23%)) and indeterminate (10/39 (26%)). The high background seroprevalence of PF4/heparin antibody (4.3–6.6%) with the preponderance of low (and frequently nonreproducible) positives in blood donors suggests the need for greater assay calibration, categorization of antibody levels and evaluation of the clinical relevance of “naturally occurring” PF4/heparin antibodies [74].

However, although anti-PF4–polyanion antibodies are common—for example, they are detected in 25 to 50% of patients after cardiovascular surgery—HIT is uncommon. Cerebral venous sinus thrombosis or thrombi in abdominal vessels rarely develop in patients with HIT. The prevalence of thrombocytopenia and anti-PF4 antibodies was studied in 492 Norwegian health care workers 11 to 35 days after Vaxzevria vaccination; anti-PF4 antibodies with optical density values over a cutoff of ≥0.4 were detected in six (1.2%) vaccines [75]. This suggests that our understanding of the pathogenesis of TTS is incomplete, and the usefulness of measuring pathogenic anti-PF4-related antibodies in all vaccine recipients is unclear. Data are needed to confirm that the anti-PF4 antibodies described here can cause thrombosis and thrombocytopenia in vivo [76].

## 9. Other Differential Diagnoses to Consider

The unusual clinical presentation of simultaneous thrombosis with thrombocytopenia, while appearing to be contradictory, is not a medical curiosity. There are a number of serious medical conditions where this can occur, as described below.

### 9.1. Microangiopathic Haemolytic Anaemia

Microangiopathic haemolytic anaemia describes non-immune (negative Coomb’s Test) haemolysis (schistocytes) and red blood cell fragmentation on a peripheral blood film and can occur in many conditions such as prosthetic heart valves and primary thrombotic microangiopathies (TMAs), including thrombotic thrombocytopenic purpura (TTP), haemolytic uraemic syndrome, complement mediated-TMA and also drug-induced-TMA [77].

Although both TTP and hemolytic uremic syndrome have many similarities, they have different aetiologies. TTP is more common in adults and is caused by severe ADAMTS13 (a Von Willebrand factor cleaving protease) deficiency, thereby invoking potent systemic thrombotic activity [78]. Only 40% of all TTP cases present with the classical ‘pentad’ of thrombosis, thrombocytopenia, fever, renal impairment and neurological symptoms [79]. By contrast, haemolytic uremic syndrome is caused by the Shiga-like toxin and mostly affects children. Renal function is invariably always impaired with less than one-third of all patients developing a fever or having neurological symptoms [79]. Primary TMA is a medical emergency with high mortality and morbidity and, as such, requires astute recognition and prompt treatment. The PLASMIC score is a validated tool using easily testable blood markers to predict the likelihood of TTP [80]. In the absence of other explainable causes whilst awaiting ADAMTS13 results, a PLASMIC score >5 warrants urgent empirical treatment such as using plasma exchange and corticosteroids. Other treatments include rituximab and anti-von Willebrand factor (caplacizumab) [81,82], usually at a specialized tertiary medical center. Other systemic disorders can also present as microangiopathic hemolytic anemia; the examples include preeclampsia and HELLP syndrome (a complication of pregnancy accompanied by hemolysis, increased liver enzymes, low platelet counts and severe hypertension), infection, solid or stem cell transplant recipients, systemic rheumatic diseases and catastrophic antiphospholipid syndrome (a rare hypercoagulative state caused by a catastrophic complement activation resulting in widespread production of microthrombi). Management of these cases requires the treatment of the underlying disorders and do not usually require TMA-specific interventions [83].

### 9.2. Disseminated Intravascular Coagulopathy

There have been unusual case reports of COVID-19 patients with ‘DIC like’ coagulopathies [84]. The mechanisms are not clearly understood but current hypotheses suggest endothelial injury related to SARS-CoV-2 invasion. DIC is a consumptive coagulopathy resulting from unregulated and abnormally activated coagulation and fibrinolysis and can range from acute to subclinical presentations. DIC always occurs in secondary underlying disorders such as infections, malignancies and severe intravascular hemolysis, as observed in an acute hemolytic transfusion reaction [85]. The generation of thrombin and the consumption of coagulation factors including platelets results in variable phenotypic expression of bleeding and/or clotting. Thus, while acute management is challenging, it is usually guided by the dominating phenotype. Unlike in TMA where the clotting profile is almost universally normal, patients with DIC present with prolonged prothrombin time and activated partial thromboplastin time with low fibrinogen levels. The severity of microangiopathic hemolysis is generally lower than TMA’s [86].

## 10. Thrombocytopenia following Vaccine Administration in Children

Thrombocytopenia is an adverse event associated with vaccine administration and can limit vaccine use due to several factors such as uncertainly about which vaccines are likely causative triggers, its incidence and severity, the risk of chronic disease and the possibility of recurrences after additional doses of the same vaccine. Vaccine-related thrombocytopenia is considered to be of immune origin because antibodies can be detected on platelets in about 79% of cases, making it a part of secondary ITPs in the subgroup of drug-induced ITPs. Thrombocytopenia following vaccine administration depends on the development of autoantibodies that cross-react with naturally present antigenic targets on platelets [87]. A comprehensive review on vaccine administrations and very rare development of ITP in children concluded that it can occur after the administration of vaccines. The only vaccine that is currently known to cause ITP is the mumps, measles and rubella (MMR) vaccine, but again the incidence of ITP is significantly lower than caused by mumps, measles and rubella, which are the diseases for which the vaccine provides 99% protection. Thus, ITP, regardless of its association with vaccination, should not limit the use of MMR vaccines, and a careful risk-benefit analysis performed particularly in children with persistent or chronic ITP should be performed. It is possible that newer technologies such as reverse vaccinology could prepare protein vaccines with a lower risk of causing ITP [88].

The role of adenoviral vectors in the development of thrombocytopenia has been described early in the pandemic. Adenoviral vectors remain ideal candidates as vaccine carriers and in cancer gene therapy due to their ability to effectively activate CD8+ T cells [89]. Early innate immune responses related to adenoviral vectors are associated with the activation of vascular endothelial cells, resulting in the release of ultra-large-molecular-weight multimers of the von Willebrand factor, a blood protein that is critical for platelet adhesion. This also activates platelets and induces the exposure of the adhesion molecule P-selectin and formation of platelet-leukocyte aggregates, ultimately causing thrombocytopenia and, thus, a risk for bleeding [90].

## 11. Post COVID-19 Syndrome and Risk of Thrombosis

Scientific and clinical evidence is evolving on the subacute and long-term effects of COVID-19, which can affect multiple organ systems [91]. Early reports suggest residual effects of SARS-CoV-2 infection, such as fatigue, dyspnea, chest pain, cognitive disturbances, arthralgia and decline in quality of life [92]. Cellular damage, a robust innate immune response with inflammatory cytokine production, and a pro-coagulant state induced by SARS-CoV-2 infection may contribute to these sequelae [93]. Long-term outcomes of patients with COVID-19 and VTE are unknown. A recent prospective study evaluated long-term bleeding, recurrence and death of COVID-19-associated VTE reported high rates of mortality (24%) and major bleeding (11%) in the first 30 days. ICU admission, thrombocytopenia and cancer indicated a poor prognosis [94].

## 12. Current Management Recommendations

The most recent updated recommendations from the Expert Hematology Panel (UK) and the American Society of Hematology suggest careful assessment of patients who present with symptoms of thrombosis 4–30 days after receiving Vaxzevria or Janssen COVID-19 vaccine [48,72]. The four diagnostic criteria below must be met:Receipt of a COVID vaccine (Janssen/Vaxzevria) 4 to 30 days previously;Thrombosis (often cerebral or abdominal);Thrombocytopenia;Positive PF4-HIT test using ELISA.

While the incidence of this thrombotic complication remains very rare, the risk of death and serious effects including thrombosis due to contracting COVID-19 nonetheless remains high. Current recommendations from both the UK and American regulators are for urgent medical evaluation for TTS if any of the following symptoms develop 4 to 30 days after vaccination: severe headache, visual changes, abdominal pain, nausea and/or vomiting, backache, shortness of breath, leg pain or swelling, petechiae or easy bruising.

If TTS is suspected, urgent diagnostic workup should be arranged, including a complete blood count with a platelet count and peripheral blood smear, imaging for thrombosis based on signs/symptoms, PF4-ELISA (HIT assay) using blood drawn prior to any therapies and fibrinogen level.

The Expert Hematology Panel (UK) classifies clinical presentations as follows:**Possible case:** Any patient presenting with acute thrombosis and new onset thrombocytopenia within 28 days of receiving COVID-19 vaccination;**Unlikely case:** Patients with either a reduced platelet count without thrombosis or with a D-dimer count at or near normal levels (<2000 µg/L) but with and normal fibrinogen (2–4 g/L) levels;**Probable case:** Elevated D-dimers (>4000 µg/L > 2000 with a strong clinical suspicion);**Definite case:** Cases usually present 5–28 days after vaccination and are characterized by thrombocytopenia, raised D-dimers and thrombosis, which often rapidly deteriorate.

There is a high preponderance of cerebral venous sinus thrombosis. Portal vein and splanchnic vein thrombosis, pulmonary embolism and arterial ischaemia are also common, as are adrenal infarction and hemorrhage. Intracranial hemorrhage can be significant and unexpected. Typical laboratory features include a low platelet count (<150 × 10^9^/L) and greatly increased D-dimer levels (above those usually expected for VTE) and many develop low fibrinogen levels contrary to hyper-fibrinogenemia observed in the acute stages of COVID-19 [95]. Antibodies associated with PF4 have been identified as in HIT but without exposure to heparin treatment. Antibodies associated with PF4 are detected by ELISA HIT assays but rarely by other HIT assay methods. The platelet count at which it is safe to initiate anticoagulation therapy is made on a case-by-case basis. The Expert Hematology Panel (UK) and American Society of Hematology revise their guidance on a regular basis as evidence emerges, with the following general recommendations: (1) Low fibrinogen or bleeding associated with TTS should not preclude anticoagulation, particularly if the platelet count is >20,000/μL or increasing following intravenous immunoglobulin initiation; (2) concurrent replacement of fibrinogen in patients with bleeding and/or very low values should be considered; (3) avoid platelet transfusions due to the similarities with HIT where platelets transfusion is relatively contraindicated unless bleeding and associated with paradoxical thrombosis. However, risk/benefit assessment in individual patients with serious bleeding and/or the need for surgical intervention may favour platelet transfusion following the initiation of intravenous immunoglobulin (IvIG), non-heparin anti-coagulation and fibrinogen replacement if level < 1.5 g/L. Platelet transfusion is an option to support therapeutic anticoagulation. However there is insufficient evidence to state that this is superior to critical care argatroban (low dose) without platelet transfusion. If urgent neurosurgery is required, then transfuse the platelets to >100 × 10^9^/L and cryoprecipitate to maintain fibrinogen >1.5 g/L [48,72].

Management involves all of the following, including probable cases while awaiting confirmatory tests:**Intravenous immunoglobulin**: Initiate urgently as this could most likely influence the disease process, using 1 g/kg (over two days if needed) irrespective of the degree of thrombocytopenia, and continue to review the clinical course. Steroids can also be helpful although it is unclear if its benefits outweigh potential harm.**Anticoagulants**: Use non-heparin-based therapies such as direct acting oral anticoagulants, fondaparinux, danaparoid or argatroban depending on the clinical presentation. Bleeding and thrombotic risk needs to be carefully assessed; low dose treatment with fondaparinux or critical illness dose argatroban may be appropriate when platelet count is <30 × 10^9^/L.**Plasma exchange**: May be considered in cases of severe or resistant disease. This may be required daily for up to 5 days if recovery is delayed.**Transfer patients with cerebral venous thrombosis**: Transfer to a neurosurgical unit and consider early recourse to neuroradiology and/or neurosurgery in cases of further deterioration/ bleeding. If urgent neurosurgery is required then transfuse platelets (to >100 × 10^9^/L) and cryoprecipitate to maintain fibrinogen levels at >1.5g/L.While it is unclear if platelet transfusion will exacerbate cerebral venous thrombosis, the risks/benefits are unknown in patients with platelets <50 × 10^9^/L on anticoagulation treatment and who have secondary cerebral bleeding not requiring procedures; therefore, clear advice cannot be given at present. Consider platelet transfusion in life threatening bleeding situations.It is unknown whether heparin exacerbates the condition but until further evidence is available, heparin is best avoided (including line flushes) as the syndrome mimics HIT with thrombosis.Replace fibrinogen to ensure levels do not drop below 1.5 g/L (using fibrinogen concentrate or cryoprecipitate where fibrinogen concentrates are not readily available).For patients who are refractory relative to repeated doses of intravenous immune globulin treatment and plasma exchange, treatment with rituximab can be considered although there is currently no evidence of its efficacy in TTS.

Further management, including intravenous immune globulin and subsequent immunomodulation, hinges on the diagnosis of thrombotic complications in the presence of thrombocytopenia driven by a positive PF4 test. TTS should also be considered in the absence of signs, symptoms or imaging documenting thrombosis with a low platelet count and greatly increased or progressively increasing D-dimer counts or a positive PF4-ELISA test. If the PF4-ELISA produces a negative result and if there is no thrombocytopenia, then TTS is ruled out. The advice is to treat for standard VTEs in such cases. Thrombocytopenia with negative PF4-ELISA in the absence of thrombosis should be managed as possible ITP.

Patients presenting with thrombosis and a normal platelet count post-vaccination might be in an early stage of TTS. These patients should be continuously assessed for tbe development of thrombocytopenia. The use of non-heparin anticoagulant may be indicated 4 to 30 days after receiving either the Vaxzevria or Janssen COVID-19 vaccine [48,72].

## 13. Evolving Clinical and Laboratories Studies

As this continues to be an evolving condition, additional data collection is underway, including serial PF4 testing, serum sample for COVID-19 antibody testing and whole genome sequencing used in tandem with standard diagnostic criteria. The addition of antiplatelet therapy to standard anticoagulation treatment may add some clinical benefits for young patients presenting with premature coronary artery disease or other arterial thrombotic conditions. As more clinical cases are being managed in the UK, argatroban monitoring has proved to be logistically cumbersome as activated partial thromboplastin time correlates poorly with benefits of argatroban (a small molecule direct thrombin inhibitor) due to high levels of Factor VIII [72]. Using argatroban can also provide false low fibrinogen levels when using some Clauss fibrinogen assays. Thus, it is recommended to use fondaparinux or another direct oral anticoagulant once bleeding risks are reduced [72].

## 14. Some Unanswered Questions about Vaccine Induced Thrombocytopenia and Thrombosis

It appears that these clots are primarily caused by autoantibodies against PF4 (currently, it is unclear why these form). These are probably IgG and can cross the blood–brain barrier, causing platelets to aggregate and collect in specific locations such as the cerebral sinus vein. ‘HIT with thrombosis’ is another rare disorder with the same constellation of findings (low platelet counts and thrombosis) that is caused by treatment with heparin but at a much higher frequency (1–5%) than TTS. Whereas HIT with thrombosis has clear risk factors, no clear risk factors have so far been identified for TTS. Heparin complexes with PF4 during HIT with thrombosis, which then attaches to platelets to trigger Fc receptor mediated platelet activation. Much about TTS still remains unclear. It appears that the mechanisms of TTS and HIT with thrombosis are quite similar in being mediated by PF4 antibodies, but clots in the brain are uncommon in HIT with thrombosis for reasons that are not clear. One possibility is that there may be differences in blood flow patterns and the size of the platelet aggregates in the two conditions.It is unclear if there is a relationship between the new variants of COVID-19 and TTS when using adenovirus vector derived recombinant vaccines.Any genetic predisposition to develop TTS has not been reported yet. Next generation sequencing may be useful in addressing this important issue.It is not clear why the cerebral circulation may be a preferred site for the formation of blood clots with some vaccines for COVID-19.

## 15. Conclusions

TTS is a dynamic target requiring urgent medical attention. Autoantibodies appear to play a critical role in the prothrombotic thrombocytopenic events that can occur during the treatment of COVID-19, particularly when using Vaxzevria and Janssen COVID-19 vaccines. National and international regulatory bodies strongly advocate for mass vaccination programs but with careful selection of patients and the avoidance of the use of Vaxzevria in patients less than 30 years of age, which may be a reasonable approach provided that alternative vaccines are available. It must be emphasized that mortality and morbidity from COVID-19 are much higher than the rare risk of developing TTS or other immune mediated complications related to the vaccines. A rapid assessment of patients presenting with the symptoms of TTS is advised, and management should be initiated without delay. It is important to report confirmed and suspected cases to regulatory bodies so a clearer picture of risk factors and vulnerable populations will emerge.

## Figures and Tables

**Figure 1 molecules-26-05004-f001:**
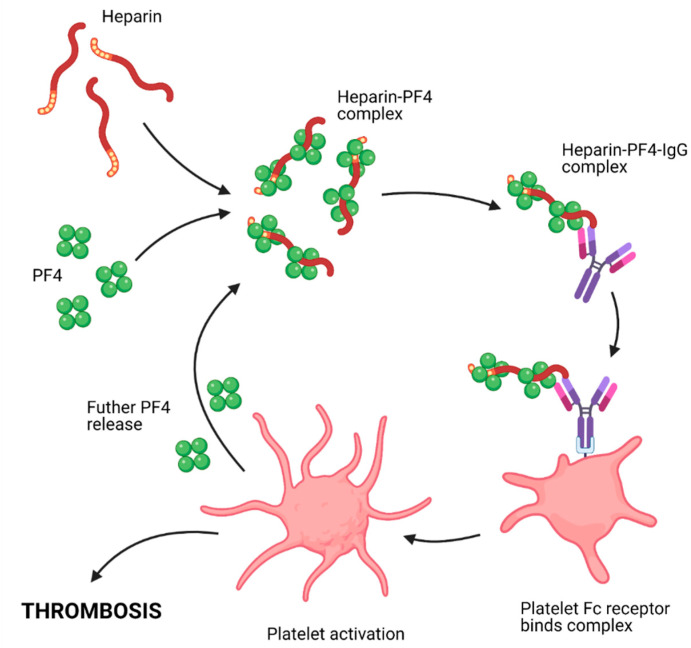
Diagrammatic representation of the mechanism by which HIT can cause thrombosis. (HIT: heparin induced thrombocytopenia.)

**Table 1 molecules-26-05004-t001:** Comparison between HIT with thrombosis and TTS.

	HIT with Thrombosis	TTS
Responsible Agent	Heparin, more likely with unfractionated rather than low molecular weight heparin. Acidic glycosaminoglycan 10–15 kilodaltons.	Vaxzevria. Adenovirus vector (approximately 150 megadaltons) or other vaccine constituent.
Onset	5–14 days following administration of heparin [63].	4–28 days following Vaxzevria vaccination [72], although majority of reported cases occurred 5–16 days following administration [61].
Presentation	Thrombosis and thrombocytopenia. Usually venous thrombosis, which can extend to unusual sites (e.g., cerebral venous sinus thrombosis, splanchnic vein thrombosis) [63].	Thrombosis and thrombocytopenia. Preponderance of unusual sites of venous thrombosis in the currently reported cases [61].
Pathophysiology	Conformational change of PF4 upon binding heparin known to be crucial to the generation of HIT antibodies and the subsequent activation of platelets [63].	Unknown.
Investigations	HIT antibodies detected with ELISA. Functional platelet assays used to confirm diagnosis [63].	HIT antibodies detected with ELISA [61].

HIT: heparin induced thrombocytopenia; TTS: thrombosis with thrombocytopenia syndrome; ELISA: enzyme-linked immunosorbent assay; PF4: platelet factor 4.

## Data Availability

No new data was created or analyzed in this study. Data sharing is not applicable to this article.

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
