# Peer review of "An Update on COVID-19 Vaccine Induced Thrombotic Thrombocytopenia Syndrome and Some Management Recommendations"

_molecules, 2021, doi:10.3390/molecules26165004_

Round 1

Reviewer 1 Report

The article presents a narrative review regarding the current state of knowledge on COVID-19 vaccine induced thrombocytopenia 2 and thrombosis as well as a review of current recommendations concerning the management of these adverse reactions.

The review is clearly structured and represents a welcome synthesis of knowledge collected from 80 referred sources. Information most relevant to the scope of this review has been extracted and has been presented in an easy-to-follow sequence and a rigorous way, using clear and understandable language.

After a brief overview and background information, the current knowledge regarding the pathophysiology of COVID-19 associated thrombosis is discussed in the beginning of this narrative review. This part of the review is followed by a synthesis of the developed vaccine types, their convincing success rates so far, as well as their serious adverse events known so far, with special attention given to vaccine induced thrombocytopenia and thrombosis (VITT). The latter is then developed in a distinct and dedicated section of the review, along with possible pathways of VITT, by similarity and contrasting with heparin induced thrombocytopenia (HIT).

The figure and table offered in this narrative review successfully participate in transmitting a synthetic and meaningful picture of the current knowledge regarding vaccine induced thrombocytopenia and thrombosis (VITT), by contrasting it with existing evidence regarding the mechanism by which heparin induced thrombocytopenia (HIT) can cause thrombosis.

Concerning the possible pathophysiology of VITT, this reviewer suggests that the authors  may  also evaluate the opportunity of including in their review a very recent hypothesis that has emerged as a preprint, concerning a possible vaccine-induced splice reaction within the SARS-CoV-2 Spike protein, that may in turn contribute to the occurrence of VITT: https://doi.org/10.21203/rs.3.rs-558954/v1

The implications and the current knowledge regarding prevalence of the involved anti-platelet factor 4 (PF4) antibodies is then given further attention in this narrative review, followed by a very well-structured and clinically useful part concerning other differential diagnoses, apart from HIT that should be considered when faced with a suspicion of VITT, including microangiopathic haemolytic anaemia, disseminated intravascular coagulopathy, thrombocytopenia following vaccine administration in children, as well as COVID-19 and post COVID-19 venous thromboembolic events (VTEs). The last 3 differential diagnoses can be deduced while lecturing the review, however, the authors may consider highlighting them more clearly, also as part of this review section concerning differential diagnoses of VITT.

A further section with important benefits for clinicians facing possible cases of VITT, is the section concerning current management recommendations, which also presents the 4 diagnostic criteria of VITT, along with the classification recommended by the UK Expert Haematology Panel, which defines unlikely, possible, probable, and definite cases of VITT. For this section, the reviewer recommends that the authors slightly alter the order in which these 4 case-categories are presented in their review, to account for the gradient of severity that underlies this classification.

The systematic presentation of diagnostic and treatment guidelines for probable cases while awaiting confirmatory tests is an important contribution of this review for a large variety of specialties that could be faced with the need to diagnose and treat the rare but potentially life-threatening cases of VITT.

Highlighting a clear set of still unresolved questions, along with the conclusions of this review, offers a good balance of evidence and still lingering uncertainties about the reviewed subject.  

The authors clearly outline the future perspectives of the reviewed knowledge, they use very clear English, and their conclusions seem well-tied and objectively linked to the reviewed evidence.

Therefore, this reviewer recommends the article to be accepted for publication, after minor changes suggested above.

Reviewer 2 Report

Line 37. Please, avoid using the word “myriad”. It sounds too belletristic. The same is for line 195.

Line 43. I believe it’s better say “summarizes the data on pathophysiology of thrombosis and thrombocytopenia in COVID-19” or something like this, than “summarizes the probable pathophysiology of COVID-19 and thrombosis”.

Line 44. Using abbreviation TTS here is not necessary as the most common term is VITT as you said before.

Lines 49-50. The common term which is used worldwide is “venous thromboembolism’ with VTE as abbreviation VTE. Please, use those two terms.

Line 87-93. The first country that approved vaccine was Russia. The vaccine was approved by regulator in Aug 2020 based on phase I/II data later published in Lancet - Logunov DY, Dolzhikova IV, Zubkova OV, Tukhvatullin AI, Shcheblyakov DV, Dzharullaeva AS, et al. (September 2020). “Safety and immunogenicity of an rAd26 and rAd5 vector-based heterologous prime-boost COVID-19 vaccine in two formulations: two open, non-randomised phase 1/2 studies from Russia”. Lancet. 396 (10255): 887–897. DOI:10.1016/S0140-6736(20)31866-3

Please, correct the information.

Lines 126-129. Such a reference as 19 seems really strange in a paper that discusses scientific data. “The Guardian” is not a proper source of data on death rates, etc. Please, exclude it. Or use the reference from scientific sources to confirm presented data.

Lines 130-136. What reference can confirm the data from Scotland?

Lines 138-145. The statements on vaccines complications presented in this paragraph need references. It’s also better to give the prevalence of different complications.

Line 193 contains reference 37 which is definitely from another version of the paper.

Section 3 and 4 look like having excessively unnecessary information sourced from media. I recommend to skip details of what different countries did regarding vaccines approving and/or suspending. The information on this is not related to  what is announced as an aim of the review.

Reviewer 3 Report

Dear colleagues,

Thanks for submitting your review to the journal. I have found a solid work that summarizes the state of the art on COVID-19 vaccination and thrombosis. Since this is not an original paper, it does not add new knowledge about this condition, as stated in the section 1.

There are several typos due to incorrect hyphenation along the first pages of text, and I would only like to mention some minor details:

Line #53: The statement of 1/5 of COVID patients with thrombosis must be contextualized to the clinically significant forms of the disease.

#71: Prophylaxis along 12 weeks after discharge. Please reference and contextualize.

#73: Incidence of thrombocytopenia. Please contrast with other sources.

#251: "The model we propose...". Maybe you meant "... we support..."?

#349: ...anti-thrombotic activity...? Please confirm.

#370: "There have been ... case reports..." Please, reference.

#381: Please, confirm that the last sentence is correct.

#461: Please, add references to the management options.

#468: "...argatraban...", please correct.

#477: Cryoprecipitate is not a valid option in many countries, where fibrinogen concentrates are used as the only source for fibrinogen.

In conclusion, I have suggested just some minor corrections to your text. I have no objections to the general structure neither the order and selection of the sections, since I understand that this is your personal approach to the review.

Regards

Reviewer 4 Report

The manuscript of Islam et al. provides an update on the thrombotic thrombocytopenia syndrome observed after vaccination with adenovirus based COVID-19 vaccines. A previous similar manuscript has been published in the literature (DOI:10.1016/j.thromres.2021.05.010) and the authors should demonstrate the added value of their manuscript to be innovative.
In addition, I have the following comments on the manuscript:
1- The authors used the term VITT which is not correct even if it has been initially used by Greynacher and reused by other colleagues. Please use the term thrombotic thrombocytopenia syndrome as used by the regulatory agencies and the Brighton collaboration. VITT suspect it concerns all vaccines which is not the case.
2- There are numerous word breaks in the document which is not appropriate.
Subsection 2: COVID-19 and thrombosis: 3: Could the authors elaborate a bit more on the role of platelets in immunoloigcla response. This could also be a factor attributed to the mild thrombocytopenia observed in patients.
4: Regarding other physiopathological mechanisms could the authors elaborate a bit more of reduced fibrinolysis (increase levels of PAI-1, probably secondary to increased levels of IL-6) 5: Contrary to TTS following vaccination with adenovirus based vaccines, fibrinogen levels are increased in COVID-19, can the authors precise this difference.
6- please update reference 10.
7- Please update the statement with Israel, there is emergence of new clusters. This is the same in other areas in the world meaning that the protection provided by vaccination is 1) either not totally efficient against the variants of concerns or 2) there is a drop in the level of neutralizing antibodies in certain populations.
8- "that occurred soon after the vaccination with Vaxzevria" please complete and be more precise (within 5 to 24 days post vaccination)
9- The authors should make a distinction between CVST and CVST associated with thrombocytopenia. This is not the same and confusion should not be made when we analyzed the yearly incidence. This should be discussed by the authors.
10- The authors should better describe the case reported in the literature because the agencies did not find any associated risk factor which permits to explain the CVST associated with thrombotcytopenia.
11- Regarding anti-PF4 antibodies, the sensitivity highly depends on the test which is used and some of them will not become positive in presence of these antibodies. This should be reflected by the manuscript (see: DOI: 10.1111/jth.15362)
12- "Indeed, the SARS-CoV-269 2 spike protein encoded by Vaxzevria induces endothelial damage through CD147-mediated cell signaling [49] as well as angiotensin converting enzyme receptor 2 dependent in need of platelet activation [50]." Reference 50 to be checked in this context and reference 49 is a preprint.
13- What is the prevalence of anti-PF4 antibodies post vaccination with Vaxzevria? (see DOI:10.1111/jth.15352)
14- In the definite case, what is the place for administrating platelet concentrates in order to have a sufficient platelet level to initiate anticoagulant therapy?
15- In the hypotheses, what about the possible activation of platelets by the adenovirus vector?
While I acknowledge that all initiatives to help further understand this adverse event is welcome, I strongly encourage the authors to update their update with the more recent literature.
Thank you for giving me the possibility to revise this manuscript.

Round 2

Reviewer 2 Report

Dear Authors!

Thank you so much for addressing all of my remarks. The paper improved a lot. 

Reviewer 4 Report

The authors correctly addressed my concerns. I have no further comments.